# communications
## engineering

PERSPECTIVE

https://doi.org/10.1038/s44172-023-00129-5　　OPEN
# Towards practical elastocaloric cooling

Yao Wang[1], Ye Liu[1], Shijie Xu[1], Guoqu Zhou[1], Jianlin Yu[1] & Suxin Qian [1✉]

Elastocaloric (eC) cooling is a promising environmental-friendly emerging cooling technology that has the potential for applications at different scales. Although the performance of eC cooling is already sufficient for some applications, a balance is needed for reliability, cost, and ease of maintenance to achieve commercialization in the near future. In this Perspective, we describe challenges and necessary steps towards practical eC cooling, including material properties and manufacturing techniques, actuators that drive eC materials, essentials for good heat transfer, and different work recovery schemes, before introducing our envisioned application scenarios.

Refrigeration consumes 20% of global energy and contributes to 7.8% of global carbon emissions[1]. In refrigeration systems, the global warming potential (GWP) of modern refrigerants themself is thousands of times worse than carbon dioxide, and without any control, the high GWP of refrigerants may contribute to >11% of global carbon emissions by 2050[2]. In addition to the continuing efforts to find environmentally friendly alternative refrigerants[3], developing zero GWP refrigeration technology also attracted tremendous attention[4–8]. Among these not-in-kind technologies, eC cooling which emerged only a decade ago outstands due to its giant cooling energy density with attractive mechanical and thermal properties[9]. As such, eC has demonstrated the potential to be engineered for applications at different scales.

eC effect refers to stress-induced heat in polymers and metals. In shape-memory alloys (SMAs) such as nickel–titanium binary alloys, other than the mechanical to thermal energy conversion, the stress-induced martensitic phase transition introduces an additional transition latent heat, resulting in colossal caloric effects, i.e. temperature change in an adiabatic unloading process can be larger than 30 K.

Any commercial product requires a perfect balance between performance, reliability, cost, and ease of maintenance. To this end, there are still many challenges to be tackled. This perspective intends to discuss these challenges towards practical application, starting from materials with recent advances in additive manufacturing that facilitate customized shape and functional-graded materials. Actuators compressing metal refrigerants ideally require large force (torque) with relatively low displacement (speed), which unfortunately are not currently off-the-shelf, and more customization and optimization are needed. Heat exchange and work recovery are the key factors determining system-level performance that compete with existing technologies. Different heat transfer enhancement methods will be discussed, as well as possible work recovery schemes. In the end, different potential application scenarios will be visited, from the traditional HVAC&R sector based on electric-driven cooling systems to heat-driven cooling applications. Since the eC material is not only refrigerant but also thermal energy storage material, application scenarios of eC as cold thermal energy storage will also be addressed.

## Solid-state elastocaloric refrigerants

The superelastic SMA determines the characteristics and performance of an eC cooling system. For practical eC cooling, an ideal SMA should exhibit large latent heat while maintaining a long fatigue life, i.e. 10 million cycles with stable and sufficient eC effect (Fig. 1a). Good mechanical properties, such as high yield stress and non-brittle, are also necessary for practical applications,

[1]Department of Refrigeration and Cryogenic Engineering, Xi'an Jiaotong University, Xi'an, China. ✉email: qiansuxin@xjtu.edu.cn

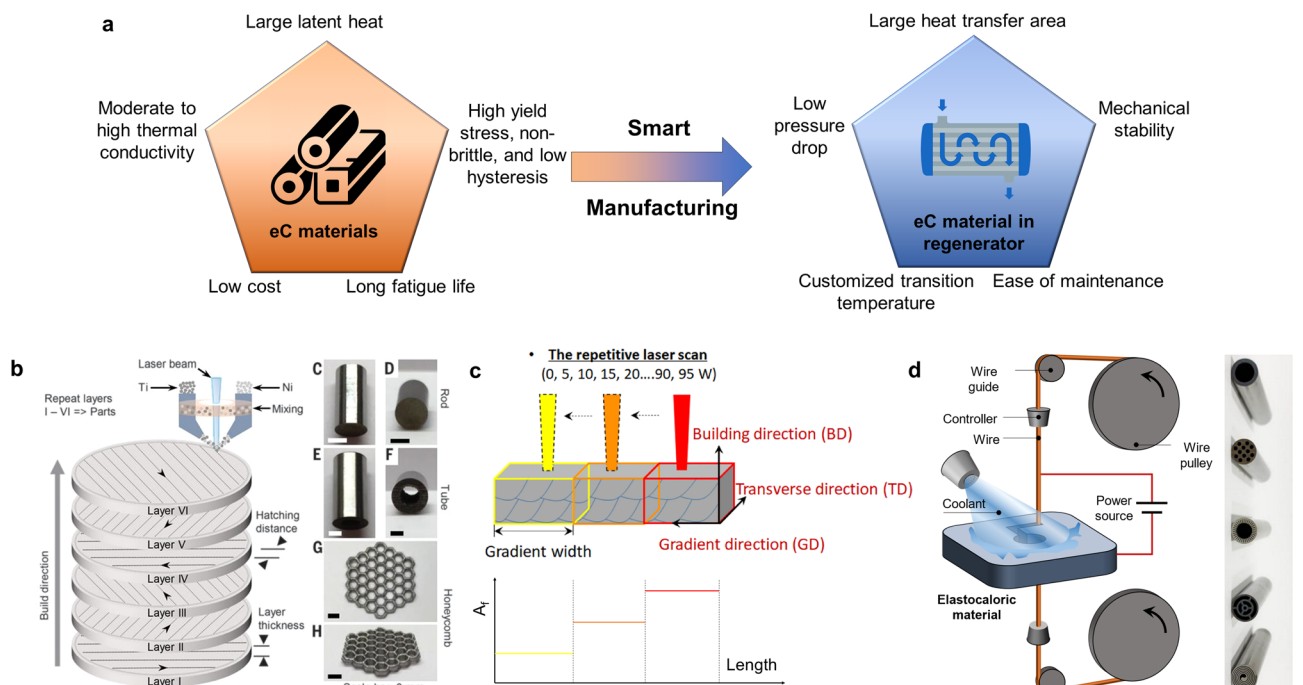

**Fig. 1 Requirements for eC materials and recent advances in eC materials manufacturing techniques. a** Requirements for an ideal eC material. **b** Additive manufacturing for customized shapes and geometries. Reprinted with permission from AAAS[19]. **c** Additive manufacturing for functional-graded eC materials with gradients of thermal properties and mechanical properties. Reprinted from Yang, Y. et al.[20] with permission from Elsevier. **d** Electric discharge machining for customized shapes and geometries[22]. Part of the image is reprinted from Zhou et al.[22] with permission from Elsevier.

in addition to a low hysteresis as a prerequisite for high efficiency. The thermal conductivity of an ideal eC material should be higher than the benchmark commercial-grade nitinol (binary NiTi alloy), but its optimum value is somewhere below $100 \, W \, m^{-1} \, K^{-1}$ to reduce the longitudinal diffusion loss in the direction of the temperature gradient of an eC regenerator with a monolithic material[10]. However, for eC regenerators that use multiple separated eC materials without diffusion loss, it is beneficial to use SMAs with even higher thermal conductivity. Furthermore, practicality comes with low cost, which means neither the raw material nor the manufacturing process should be too expensive for highly competitive commercial products such as home appliances.

So far, commercial-grade nitinol is the only option for almost all eC cooling prototypes published to date, with a few exceptions using elastocaloric polymers[11,12]. Nitinol features decent latent heat ($10–15 \, J \, g^{-1}$), equivalent to 20–30 K adiabatic temperature change, with remarkable mechanical properties and acceptable fatigue performance under compression. Taking thermoelectric refrigerators as an example, a state-of-the-art thermoelectric module produces a 70–80 K maximum temperature span and can be operated at a 30–40 K temperature span with dozens of wattage as useful cooling. For eC materials, to deliver useful cooling at a similar temperature span, together with higher and more competitive efficiency than thermoelectric, the adiabatic temperature change (and latent heat) of the benchmark commercial-grade nitinol is most likely already sufficient since active regeneration can further boost the system temperature span by a factor of two or more (two for eC[13] and more than five for electrocaloric regenerator[14]). However, the hysteresis of commercial-grade nitinol is too large in terms of efficiency to be competitive with vapor compression. As such, developing eC materials with low hysteresis and long fatigue life, such as ternary TiNiCu or TiNiCo alloys are important (Table 1). Although compression is better than tension in terms of fatigue life[15], it is also important to

develop fatigue-resistance materials under tension, because fluid-free eC cooling systems driven by tension can be more versatile for small-scale applications[16]. Materials with excessive latent heat but limited mechanical or fatigue performance, such as NiMnTiB, may be more suitable for eC thermal energy storage applications, which will be discussed in the last section.

Manufacturing raw eC materials into porous structures with desired shapes and properties is also important. An ideal eC regenerator features a large heat transfer area with a low-pressure drop (Fig. 1a). The eC regenerator should be free from buckling if compression is applied. Advanced functionality, such as customized distribution of transition temperature, is needed for some of the eC regenerator geometries, to mitigate inhomogeneous strain distribution and produce a more uniform eC effect across the eC regenerator[17,18]. To achieve these goals, smart manufacturing techniques are needed. Additive manufacturing, a.k.a. 3-D printing, has been demonstrated to be an effective way to produce eC materials with customized shapes and functionality. For example, the laser-directed-energy-deposition (L-DED) technique was used to melt Ni and Ti powders, where the moving molten pool was rapidly cooled to form customized shapes layer by layer (Fig. 1b)[19]. Different shapes, such as rods, tubes, and honeycomb structures, were printed and tested. In addition, when the feeding flow rate of Ni and Ti powders is finely controlled, the same technique can produce a variable-composition structure, such that a gradient of transition temperature is formed. A different additive manufacturing technique is selective laser melting, where a bed of Ni and Ti powders is melted by laser with prescribed input power and route (Fig. 1c)[20]. More specifically, the input power of the laser is controlled by the number of repetitive pulses, because the vaporization of Ni occurs most effectively in the first few milliseconds upon application of laser[21]. By tuning the input pulses and power, different segments along the gradient direction (Fig. 1c) can be built with different compositions and transition temperatures.

**Table 1 Performance of selected eC materials.**

| Materials | Latent heat [J g$^{-1}$] | Thermal conductivity [W m$^{-1}$ K$^{-1}$] | Transition stress [MPa] | Isothermal hysteresis [J g$^{-1}$] |
|---|---|---|---|---|
| Commercial grade NiTi[55] | 12 | 18 | 700 | 0.46 |
| TiNiCu[56] | 8 | 40 | 300 | 0.2 |
| NiFeGaCo[57] | 6 | 17 | 130 | 0.34 |
| CuZnAl[58] | 5 | NA | 250 | 0.12 |
| NiMnTiB[59,60] | 13–50$^a$ | 18 | 700 | 0.71 |

$^a$50 J g$^{-1}$ at 300 °C not room temperature[60].

On the other hand, if controlled properly, traditional subtractive manufacturing can be also powerful enough to produce eC materials with customized shapes. Recently, the research team at Hong Kong University of Science and Technology demonstrated multiple novel tubular eC regenerator geometries that were fabricated by electric discharge machining (EDM) (Fig. 1d)[22]. Customized flow channels were formed using EDM wire with a fine diameter, which features a large heat transfer area while being resistant to buckling. This approach may be combined with the aforementioned two additive manufacturing techniques, where a more precise control of customized shape is handled by traditional manufacturing such as EDM, while the customized composition for variable transition temperature is achieved by additive manufacturing.

**Actuators for compressing metals**
In classical thermodynamics, solids are considered incompressible. With the martensitic transition, SMAs become stretchable and compressible. However, the strain variation of SMAs is magnitudes less than the volumetric change in gases. Therefore, compared with actuators that compress gases, actuators that compress SMAs feature much less displacement. As compensation for a fixed input work, in Eq. (1), the driving stress becomes much larger. Thus, an ideal actuator for eC cooling application requires a higher force and a lower displacement (stroke), when compared with off-the-shelf actuators that are optimized for a balance between force and displacement (Fig. 2a).

$$W = F \cdot \Delta x \qquad (1)$$

where $W$, $F$, and $\Delta x$ denote the work [J], the force of actuators [N], and the displacement of actuators [m], respectively.

To produce 1 kW cooling power, or equivalently 500 W at half of the zero-load temperature span, 200 g of eC materials are needed since current eC regenerators already featured a cooling power density of around 5 W g$^{-1}$ [13]. Using the properties of nitinol as an example, and using the length from the multi-mode eC prototype[23], 123 mm$^2$ cross-sectional area is needed, which translates into 98.5 kN loading force assuming 800 MPa transition stress. Meanwhile, 10 mm displacement is needed for a 4% strain. Therefore, these values are set as the coordinate to compare commercially available actuators in Fig. 2b.

Hydraulic actuators are well-known for their high output force and have been adopted in recently published prototypes[13,23]. Ideally, the stroke of a hydraulic actuator can be customized to applications. However, the stroke of the in-stock models is often significantly more than the target, and thus, unless fully customized, the footprint of hydraulic actuators is usually oversized. Thus, it is important to further customize and optimize compact hydraulic actuators (cylinders) for large-scale eC cooling applications. Furthermore, designers must understand that an additional hydraulic loop comes with the actuators, including a hydraulic oil tank, a pump, valves, and pressure gauges. Besides, the efficiency of future hydraulic actuators must be improved,

since in the state-of-the-art open-loop design, the high-pressure oil is discharged to the tank at zero pressure, wasting a significant amount of input work from the hydraulic pump, in addition to the relatively low efficiency (30–60%) of these hydraulic pumps. Although the work recovery design reduces such losses in our multi-mode eC cooling system[23], it is important to use a closed-loop design to further reduce the wasted pressure differential in the hydraulic oil for higher efficiency. Using screws or other mechanisms, linear actuation can be achieved using conventional electromechanical motors, which have been the most widely used actuator since the first eC cooling prototype[16,24–30]. Most commercially available electromechanical linear actuators have a built-in gearbox to increase torque, except for a customized high-torque motor that directly drives a screw slider for improved compactness[16]. In the future, for applications with multiple units of eC regenerators, electromechanical motors with the crankshaft mechanism may become a viable option.

The actuator that best matches the eC cooling application in terms of output characteristics, as well as energy density, is the SMA actuator[31,32]. The major drawback of an SMA actuator is efficiency, followed by actuating speed. The low efficiency is because the state-of-the-art electricity-driven SMA actuators reject most of the heat to ambient without any heat recovery. Our group proposed a regenerative SMA actuator scheme that recovers such previously wasted heat based on fluid heat exchange[33,34]. Reducing the heat input by higher efficiency also leads to a much shorter time to complete actuation. Unlike the commercially available technologies, more efforts are needed to further investigate and optimize the regenerative SMA actuator.

Other actuating technologies may have the potential for eC cooling, but they all have intrinsic limitations. For example, there is one eC cooling demo using magnetostriction as the driver[35], but the magnetostrictive actuator is too oversized to compensate for its limited stroke. This also applies to piezoelectric actuators. Solenoid actuators are widely used for control applications, for example, controlling the state of valves and relays, because of their limited actuating force.

**Heat exchange and work recovery**
Heat exchange plays the most important role in the performance of any eC cooling system. Improving the heat exchange is beneficial as long as the thermodynamic cycling properties are not compromised. To improve heat exchange with a fixed heat transfer temperature difference, both the enhanced heat transfer coefficient and the increased heat transfer area are needed (Eq. (2)).

$$Q_v = h \cdot \beta \cdot \left( T_f - T_{SMA} \right) \qquad (2)$$

where $Q_v$ is the volumetric cooling power [W m$^{-3}$]. $h$ and $\beta$ denote the heat transfer coefficient between the SMA and the heat exchange medium [W m$^{-2}$ K$^{-1}$] and the specific heat transfer area (heat transfer area per volume) [m$^2$ m$^{-3}$], respectively. $T_f - T_{SMA}$ is the heat transfer temperature difference [K].

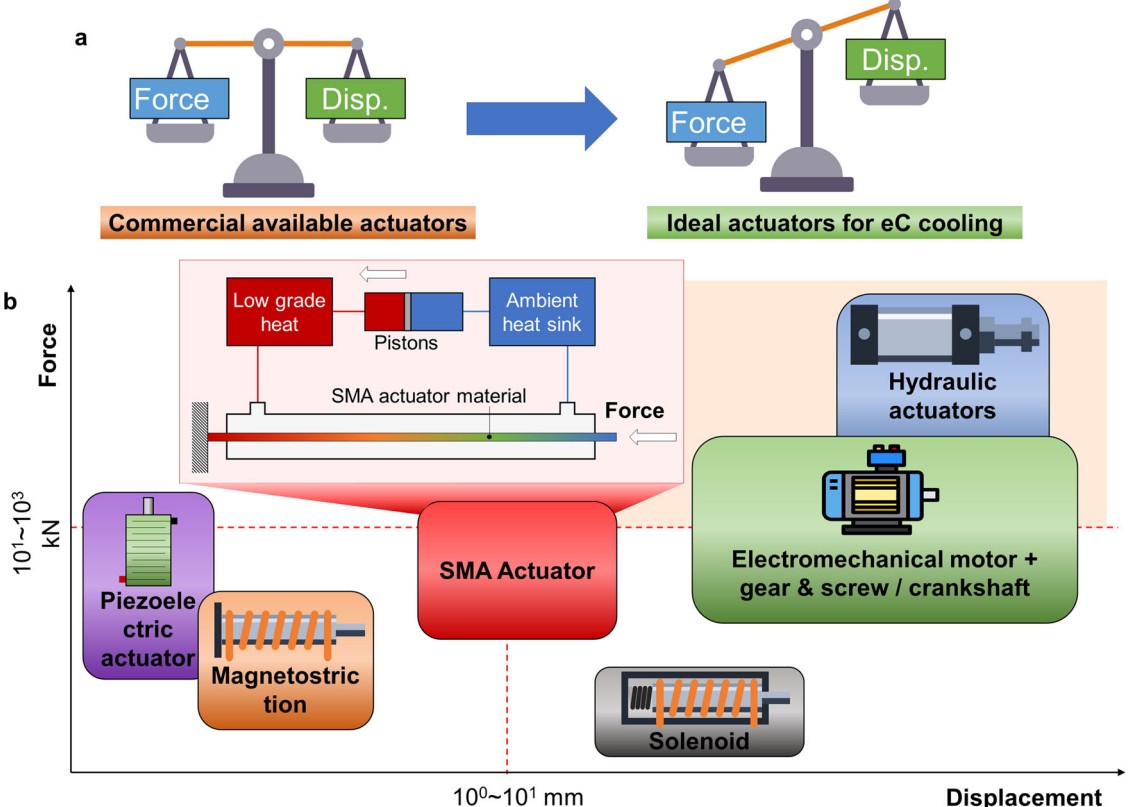

**Fig. 2 Actuators for eC cooling. a** Requirement for an ideal actuator in eC cooling systems. **b** Comparison of different actuators (data based on refs. [32,65]). The dashed lines mark the force and displacement requirements for a typical eC cooling system. The color of the SMA actuator represents temperature, i.e. blue for low temperature and red for high temperature, following the traditional rainbow color map. Other background colors do not have any physical interpretation. Hydraulic actuators and electromechanical motors have been adopted in existing eC cooling prototypes while SMA actuators are still under development. Magnetostriction has been demonstrated in a lab-scale device. Piezoelectric and solenoid actuators have major limitations in displacement and force, respectively.

The first approach is increasing the specific heat transfer area, $\beta$. The baseline for comparison here is single-phase forced convection inside an SMA tube[26]. A bare tube is listed as the baseline, not only because it was applied in the first compressive eC cooling system, but also because of its easeness of manufacturing and availability from the market. Inserting more SMA tubes or rods inside an SMA tube with a larger diameter may increase the heat transfer area[36]. Recently, the research team at Hong Kong University of Science and Technology proposed multiple novel geometries with significantly larger heat transfer areas using the EDM technique, including the microchannel structure, micro-fin structure, and spiral structure (Fig. 3a)[22,37]. Among these structures, the spiral structure with 0.15 mm wide flow channels features the largest heat transfer area, followed by the micro-fin structure. Furthermore, the spiral structure features a uniform thickness of the SMA wall, which is preferable for mechanical stability. Other than tubular geometries, thin parallel-plate geometry was proposed for the eC regenerator that also features a large heat transfer area[38], albeit cannot be driven by compression. Buckling-resistance geometries such as wavy plates may be an alternative, which also increases the heat transfer coefficient, $h$, because it periodically interrupts the flow boundary layer[39].

The second approach is increasing the heat transfer coefficient, a.k.a. heat transfer enhancement. In eC regenerators, the flow velocity is usually <1 m s$^{-1}$, and considering the geometries, the flow regime is usually laminar[40]. For single-phase laminar flow, the Nusselt number is independent of the Reynolds number[41]. As a result, reducing the hydraulic diameter is effective to increase

the heat transfer coefficient, since the temperature gradient for diffusion becomes larger. In fact, reducing the diameter of tubes is an ongoing effort in mainstream air-conditioning and refrigeration devices[42]. Alternatively, for the same tube diameter, adding insertions can also increase the temperature gradient in the transverse direction and thus increase the heat transfer coefficient, which was proposed for bare tubes first[23,36], and was recently implemented for tubular structures with more complicated geometries[22,37]. In addition, compared with circular tubes at the same hydraulic diameter (and surface area), the honeycomb structure features a higher heat transfer coefficient, because of a more confined flow that promotes convection (Fig. 3a).

The aforementioned discussions are mainly focused on internal flow. For bare SMA tubes with the same diameter, external flow around tube banks generates turbulences, facilitating a higher heat transfer coefficient than internal flow. Taking water flowing at 0.2 m s$^{-1}$ at 25 °C as an example, the convective heat transfer coefficient for flow inside a 3 mm bare tube (internal diameter) is 740 W m$^{-2}$ K$^{-1}$, while the convective heat transfer coefficient for an external flow of the same outer diameter with 0.4 mm tube spacing is 4300 W m$^{-2}$ K$^{-1}$, representing a fivefold improvement. As such, eC regenerators with this external flow were developed with decent cooling power density[13,43–45]. The heat transfer coefficient can be further enhanced when the single-phase convection is replaced by two-phase convection, i.e. convective boiling and condensation[46]. However, how to achieve active regeneration with a large temperature gradient along the flow direction using the isothermal two-phase convection remains a challenge for further investigations.

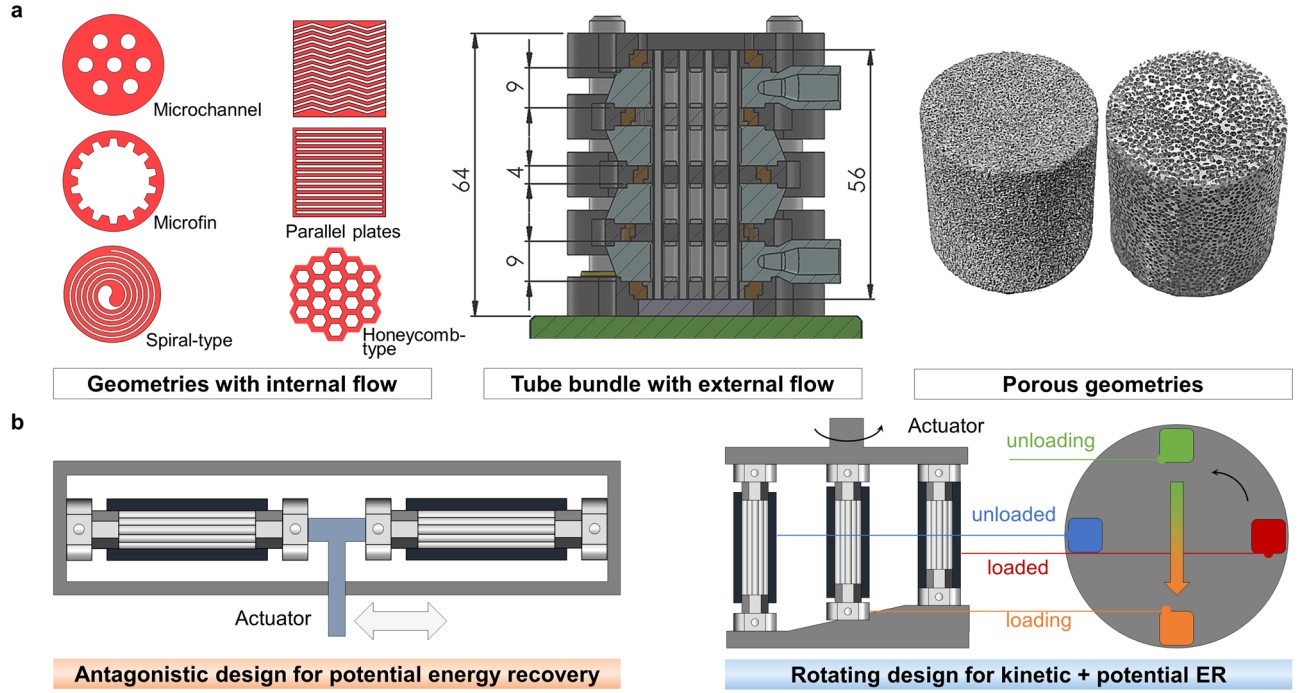

**Fig. 3 Opportunities in enhanced heat transfer and advanced work recovery. a** Summary of geometries with heat transfer enhancement in eC cooling. Part of the image is reprinted from Ahčin, Ž. et al.[13], with permission from Elsevier. **b** Concepts for work recovery (ER means energy recovery). The color of the eC regenerators in the rotating design schematic represents temperature, i.e. blue for low temperature and red for high temperature, following the traditional rainbow color map. In the rotating design, the eC regenerators are rotated counterclockwise continuously, such that the potential energy from the unloaded eC regenerator assists the loading of the opposite eC regenerator while its kinetic energy is conserved instead of wasted in the antagonistic design.

Other than the ordered geometries, disordered geometries, such as foam SMAs, were recently theoretically studied[47]. The advantage of foam structure is a much higher specific heat transfer area, $\beta$. A higher heat transfer coefficient can also be expected for foam with an average pore size of 0.2 mm[47]. The major drawbacks are much higher pressure drop, inhomogeneous stress distribution, and short fatigue life compared with solid eC materials. Other disordered geometries should also be visited in the future.

Heat transfer determines the cycling frequency and cooling power, while work recovery directly reduces the input power and increases efficiency. For eC cooling applications, work recovery consists of potential work recovery and kinetic work recovery[48]. In the antagonistic configuration (Fig. 3b)[23,26,27], two eC materials are pre-compressed to 50% of the maximum strain and are operated half cycle out of phase from each other, i.e. when one eC material is loaded, the other one is unloaded. At the very beginning of the work recovery process, the force from the unloading eC materials is significantly larger than the required force for the eC materials being loaded. As a result of such an imbalance of force, a significant part of the potential energy that is stored in the compressed eC materials converts to kinetic energy during its unloading process. This kinetic energy is wasted and irrecoverable in the antagonistic design because the eC materials decelerate and eventually become static for heat transfer. The remaining potential energy assists the loading of the other eC materials by reducing the input force of the actuator. Hence, the work recovery efficiency of the antagonistic configuration is <100%, which was estimated to be around 77% using the stress–strain response of a commercial-grade NiTi material[48].

To recover both the potential energy and the kinetic energy, one option is to keep a constant rotation for eC materials throughout the entire cycle (Fig. 3b). In the rotating concept, the eC materials are attached to a rotating top disk. The bottom static disk features a profile such that half of the system contains eC materials that are either fully loaded or being loaded, where a stream of fluid rejects the heat from the eC materials. The other half of the system contains unloaded or unloading eC materials that provide cooling to a second stream of fluid. In this design, since the top disk and all eC materials are always rotating at a constant speed, no variation of kinetic energy is needed, except for the initial start-up process. Consequently, the kinetic energy from the unloading eC materials can be fully recovered to reduce the torque of the actuator. Several prototypes adopted this design[49,50], including the very first eC cooling demo at the University of Maryland[51]. Despite the potentially high work recovery efficiency, rotating multiple eC materials that require hundreds of megapascals of stress often leads to huge friction. If not mitigated properly, the friction can easily outbalance the benefit of higher work recovery efficiencies.

Considering eC cooling already achieved more than 50 K temperature span and hundreds of watts of cooling[22], from now on, it becomes more and more important to pursue higher efficiency. Other than developing more efficient eC materials and actuators, work recovery is an important feature that must be included in practical eC cooling systems, especially for eC materials with higher efficiency since their materials' efficiency will drop more significantly without work recovery[52].

## Opportunities and potential application scenarios

As revealed by a recent study, the performance of eC cooling increases significantly in the past few years among all emerging caloric cooling technologies[23]. Since eC cooling can already produce a temperature span at zero loads ($\Delta T_{\text{span,max}}$) of more than 50 K and deliver more than 200 W cooling at zero

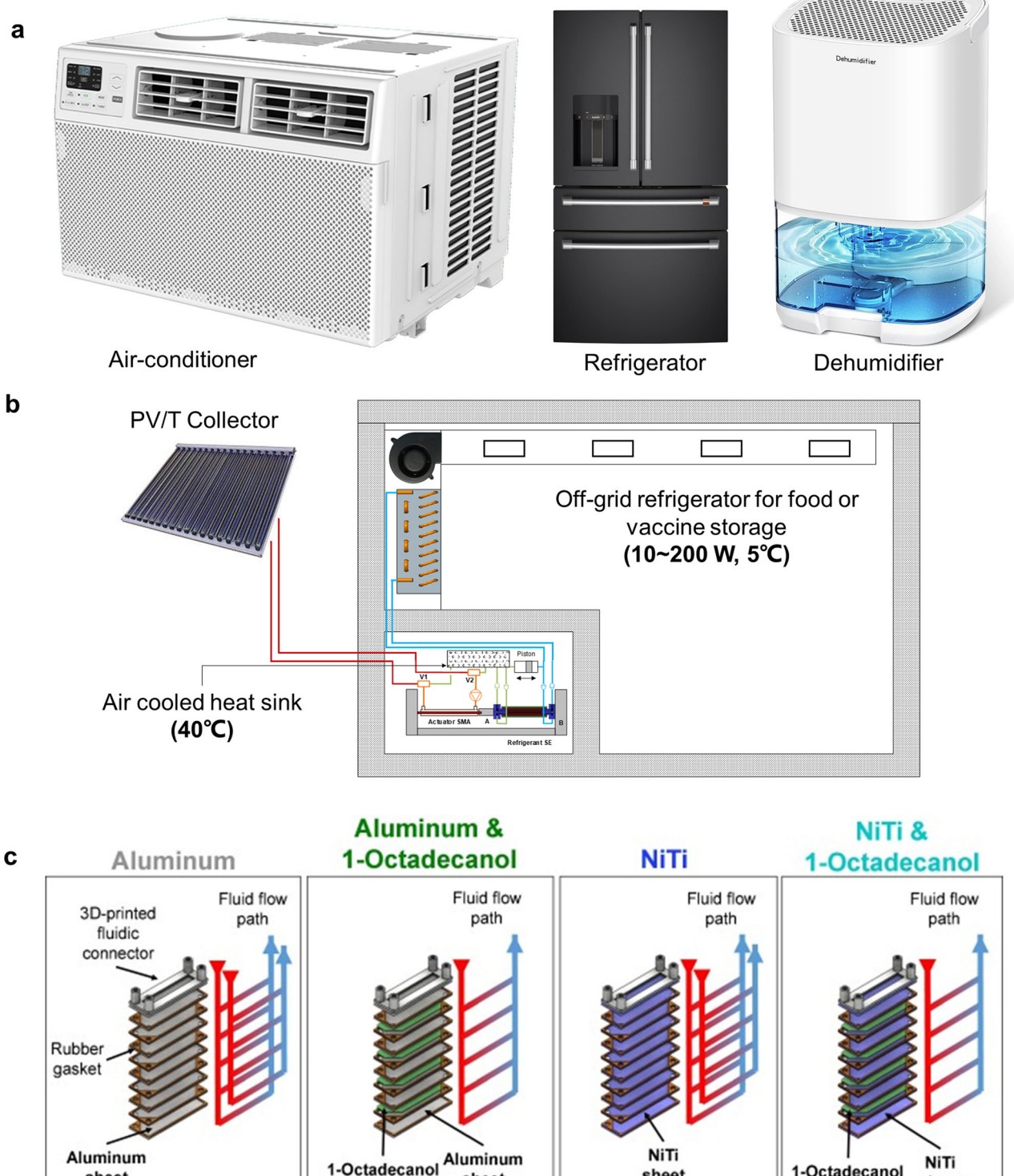

**Fig. 4 Potential application scenarios of eC cooling technology. a** Mainstream HVAC&R applications including electric-driven eC water chillers for air-conditioners, refrigerators, and dehumidifiers. **b** Heat-driven applications for waste heat recovery and off-grid refrigeration. The color of the heat transfer fluid pipelines represents temperature, i.e. blue for low temperature and red for high temperature, following the traditional rainbow color map. Reprinted from Qian, S. et al.[33], with permission from Elsevier. **c** eC-based thermal energy storage applications. Reprinted from Sharar et al.[54], with permission from Elsevier.

temperature span[22], it is reasonable to be optimistic that in a few years, eC cooling can meet the performance requirements for small refrigeration equipment (<10 kW), such as air-to-air unitary air conditioners and heat pumps, domestic refrigerators, and dehumidifiers (Fig. 4a).

Taking a window-type air conditioner with 2.5 kW nominal cooling power as an example, the 2.5 kW capacity at rated condition implies 5.0 kW cooling at zero temperature span and 59 K maximum temperature span at zero capacity (Table 2), assuming a linear correlation between the cooling power and temperature

**Table 2 Rating conditions for commercial refrigeration products and performance requirements for practical eC cooling systems.**

| Category | $T_c$ [°C] | $T_h$ [°C] | Cooling (heating) power and corresponding $\Delta T_{span}$[a] | $\Delta T_{span,max}$ at 0 load [K][b] |
|---|---|---|---|---|
| Split-type air-conditioner (cooling)[61] | 15.5[c] | 35 | 2.5 kW at 29.5 K | 59 |
| Split-type air-conditioner (heat pump)[62] | −7 | 20 | 3.0 kW at 37 K | 74 |
| Domestic refrigerator (fresh food)[63] | 4 | 32 | 100 W at 38 K | 76 |
| Dehumidifier[64] | 23.7[d] | 27 | 1 kW at 13.3 K | 26.6 |

[a]Assuming 5 K heat transfer temperature difference on each side.
[b]Assuming a linear characteristic curve between temperature span and cooling. (heating) power, which implies that the maximum cooling (heating) power is twice the value listed in the table.
[c]Dew point temperature corresponding to the standard dry bulb of 27 °C and wet bulb of 19 °C.
[d]Dew point temperature corresponding to the standard dry bulb of 27 °C and wet bulb of 24 °C.

span on the performance map that has been previously reported[13,22]. Cost is the primary concern for any potential customer for eC cooling technology, which scales primarily with weight. With advances in heat transfer, it is reasonable to assume that 10 W g$^{-1}$ specific cooling power can be achieved in the future. In this case, 500 g of SMA is needed. Assuming 250 mm length for two antagonistic eC regenerators, the requirement force and displacement for the actuator are 46 kN and 10 mm, respectively, if ternary TiNiCu with transition stress at 300 MPa is available for mass production. Off-the-shelf electromechanical actuator with 50 kN capacity is usually more than 15 kg. More importantly, the weight of frames that could sustain the 46 kN load should also be accounted for. Optimum design for the frames and customized compact actuators without unnecessary yet expensive control features and excessive accuracies are needed to compete with the weight and cost of compressors in state-of-the-art window-type air conditioners.

Despite the mandatory requirement for cooling power and temperature span, to compete with vapor compression technology for small refrigeration equipment (Table 2), efficiency is the most important factor other than cost. In a kW capacity eC cooling system, a pump and multiple solenoid control valves are needed. If scaled properly, the pressure drop across eC regenerators should be <0.3 MPa[22], and for 500 g for SMA materials, the pumping power is trivial, i.e. <15 W, assuming 1 Hz operating frequency (0.25 s for heat transfer in each half cycle), the porosity of 0.2, utilization factor of 0.5, and pumping efficiency of 30%. The majority of input power depends on the actuator efficiency, the work recovery efficiency, and ultimately, the material hysteresis. As a reference, the total input power for a 2.5 kW capacity air conditioner is <735 W.

Other than the mainstream electric-driven cooling applications, the heat-driven SMA actuator facilitates off-grid cooling applications, e.g. a vaccine refrigerator in off-grid regions (Fig. 4b)[33]. The photovoltaic/photothermal (PV/T) collector provides power for a controller and circulating pump, while one unit of input thermal energy at 110 °C translates to 0.8 units of cooling at zero temperature span or 0.14 units of cooling at 38 K temperature span. When providing 200 W cooling to a vaccine refrigerator with a glass door (300–500 L storage volume), a PV/T collector with 1.5 m$^2$ area is needed, in addition to 0.85 kg of SMA for the actuator and refrigerant[33]. Nonetheless, much more effort is needed to further validate the theoretical performance and application potential of heat-driven eC cooling.

SMA materials are not only refrigerants but also good candidates for thermal energy storage. This dual functionality of SMA is superior to conventional vapor compression refrigerants, where a dedicated thermal energy storage material is needed, such as ice or paraffin wax. For specific applications where cooling (or heating) is only occasionally needed, eC cooling integrated with the thermal energy storage concept is a better solution, where the actuator with a much smaller capacity can gradually load multiple eC materials slowly (isothermally) while maintaining their loaded state. When cooling is needed, all eC materials discharge their cooling capacity instantly[53]. Alternatively, even without stress, eC materials with $A_f$ between ambient temperature and the target cooling temperature (usually for electronics) can use their martensite to austenite latent heat during discharging and resume the cooling capacity afterward (Fig. 4c). Although the mass-based latent heat of SMA is still limited, the volumetric thermal energy performance of SMA outperforms traditional paraffin-based materials, manifested by a much higher thermal conductivity in SMA than paraffin and its composite structure with copper or aluminum[54]. Consequently, for applications where volume rather than weight is a limiting factor, eC-based thermal energy storage may provide a unique solution. Similar to heat-driven eC cooling, thermal energy storage by eC is a new concept, and more studies are needed for system-level integration and validation.

## Future perspective
Over 15 million people worldwide are estimated to work in the field of vapor compression refrigeration technology which has evolved for more than two centuries, and it was the invention of chlorofluorocarbons in the 1920s that facilitated the domination of vapor compression over vapor absorption technology. eC cooling was born merely a decade ago with only dozens of researchers and yet has demonstrated huge potential as a next-generation cooling technology. Much more joint efforts from industry and academia are needed to overcome the multi-disciplinary challenges from material research to engineering practice and eventually to commercialization, where the academia explores new materials, cutting-edge materials manufacturing techniques, and novel eC regenerator architectures, while much more efforts from industry are needed in terms of scaling up the materials manufacturing capability and providing specialized actuators with large force at low cost.

## Data availability
All data are available in the manuscript.

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

## Acknowledgements

This work was financially supported by the National Natural Science Foundation of China (NSFC) under grant no. 51976149.

## Author contributions

S.Q. and Y.W. conceived the concept. J.Y. and S.Q. supervised the research. S.Q., Y.W., Y.L., S.X., and G.Z. prepared the graphics. Y.W. and S.Q. conducted data analysis. S.Q. and Y.W. wrote and edited the manuscript with substantial input from the other authors.

## Competing interests

The authors declare no competing interests.
