## [Peer Review File · Communications Engineering]

Reviewers' comments:

Reviewer #1 (Remarks to the Author):

Dear Authors; dear Editor;

The manuscripts give a good overview of the current status of the elastocaloric technology and, more importantly, it gives a view on the future requirements that need to be reached and solved in order to make elastocaloric technology commercially viable. The manuscript is well written, but the following minor comments must be addressed before publication.

1. In line 46 diffusion losses are discussed. I would recommend that the authors rather talk about longitudinal heat conduction losses along the eC regenerator in the direction of established temperature profile.
2. It is stated in line 56 that thermoelectric module produces 70-80 K of adiabatic temperature span. I think the word "adiabatic" is redundant.
3. It is stated in line 62 that hysteresis in commercial Nitinol is too large to be competitive with vapor compression. I think the authors mean in terms of efficiency.
4. I would suggest that the authors briefly mention also elastocaloric polymers in the manuscript.
5. It is mentioned (line 115) that actuator for ec cooling requires high force (torque). Indeed, high force is required, but the torque largely depends on the system design and lever on which the eC material is.
6. In line 123 the required force is calculated. However, it should be noted that this force is for a single eC regenerator, but if some force recovery mechanism is applied, the required input force and particularly the torque can be significantly reduced.
7. It is stated in line 134 that the efficiency of hydraulic actuators must be improved. What is the typical efficiency of such actuators?
8. The authors stated that work recovery reduced losses in multi-mode eC system (line 137). Did the authors mean multi-regenerator instead of multi-mode?
9. In line 216-220 disordered geometries are mentioned. An important disadvantage of such design is also highly inhomogeneous stress distribution, which further results in inhomogeneous eCE and thus poorer performance. This should be mentioned.
10. It is mentioned in line 239 that in rotary systems air is usually used. I do not think it is necessary to use air, as also water can be used in a similar way. I would suggest omitting "(usually air)" from that sentence.
11. It is not clear why window-type air conditioning device require maximum temperature span of 59 K, if it usually work with a temperature span of about 20 K. this should be explained more in detail.
12. It is mentioned (line 281) that the total input of air conditioner is less than 735 W. At which temperature span?
13. In the last paragraph SMA thermal storage is discussed. Indeed, SMA has many advantages compared to traditional PCMs, but their latent heat is still quite smaller, which should be

mentioned in the manuscript.

Reviewer #2 (Remarks to the Author):

This is a review of the application research of elastocaloric cooling. The manuscript provides detailed state-of-the-art elastocaloric cooling and forecasts for the development of this technology. It helps to understand the elastocaloric application and inspires the researchers in this field. There are some minor shortcomings to be addressed to improve this review article.

1. The title of each section should be enlightened. Obviously, the present 2nd and 3rd paragraphs are not included in the Abstract. They should be in the section "Introduction". At the end of the Section "Introduction", the main contents or structure of this manuscript should be depicted clearly.

2. At present, the already appeared prototypes of elastocaloric cooling almost used NiTi alloy. So, the title of this article should add "NiTi alloy". If you present other materials of elastocaloric application, in the text, they should be listed and the comparison in the coefficient of performance (COP), stress hysteresis, or anti-fatigue performance between them is necessary.

Response to Reviewer Comments

Dear editor & reviewers,

We would like to thank you for your careful reading, helpful comments, and constructive suggestions on our manuscript entitled “Towards practical elastocaloric cooling”. We appreciate the time and effort you took to review our work and provide valuable feedback.

We have carefully considered and addressed all of your comments and suggestions. Below please find our responses.

Reviewer #1

Dear Authors; dear Editor;

The manuscripts give a good overview of the current status of the elastocaloric technology and, more importantly, it gives a view on the future requirements that need to be reached and solved in order to make elastocaloric technology commercially viable. The manuscript is well written, but the following minor comments must be addressed before publication.

Comment 1-1 In line 46 diffusion losses are discussed. I would recommend that the authors rather talk about longitudinal heat conduction losses along the eC regenerator in the direction of established temperature profile.

Response: Thank you very much for your suggestion. The diffusion loss has been updated to “longitudinal diffusion loss in the direction of the temperature gradient of an eC regenerator” according to your suggestion.

Comment 1-2 It is stated in line 56 that thermoelectric module produces 70-80 K of adiabatic temperature span. I think the word “adiabatic” is redundant.

Response: Thank you very much for your suggestion. The adiabatic refers to no load condition. To avoid confusion, the “adiabatic temperature span” has been updated to “maximum temperature span”.

Comment 1-3 It is stated in line 62 that hysteresis in commercial Nitinol is too large to be competitive with vapor compression. I think the authors mean in terms of efficiency.

Response: Thank you very much for your suggestion. “in terms of efficiency” has been added accordingly.

Comment 1-4 I would suggest that the authors briefly mention also elastocaloric polymers in the manuscript.

Response: Thank you very much for your suggestion. We added “...with a few exceptions using elastocaloric polymers^{11,12}” in the first sentence of the second paragraph in the materials section with two additional references on elastocaloric polymers.

Comment 1-5 It is mentioned (line 115) that actuator for ec cooling requires high force (torque). Indeed, high force is required, but the torque largely depends on the system design and lever on which the eC material is.

Response: Thank you very much for your suggestion. We agree. The “(torque)” was removed in the revised version.

Comment 1-6 In line 123 the required force is calculated. However, it should be noted that this force is for a single eC regenerator, but if some force recovery mechanism is applied, the required input force and particularly the torque can be significantly reduced.

Response: Thank you very much for your comment. Unfortunately, we cannot concur with you on this. The input work and the input force during the loading is indeed reduced with work recovery, but the maximum input force is still the same. This is because when an eC regenerator is fully unloaded, it cannot provide any unloading force to the other regenerator being loaded (see Fig. 8 in ref. 53 for detailed discussions). As a result, when sizing the actuator, the maximum stress/force should be used as the criterion.

Comment 1-7 It is stated in line 134 that the efficiency of hydraulic actuators must be improved. What is the typical efficiency of such actuators?

Response: Thank you very much for your comment. The typical efficiency of hydraulic pumps is 30~60%, according to D. R. Grandall, thesis, University of Minnesota (2010). This efficiency evaluates the output mechanical work at a given input electric power. However, a significant portion of the output mechanical work from the pump is not used by the actuator but wasted, because during the unloading process, the high-pressure hydraulic oil is discharged to ambient pressure, which requires re-pressurization in the next loading cycle. Thus, although it is difficult to quantitatively describe the efficiency of hydraulic actuators at this moment, we know there are plenty rooms for improvements.

In the revised version, we added “..., in addition to the relatively low efficiency (30~60%) of these hydraulic pumps” to complement the statement of “the efficiency of future hydraulic actuators must be improved, since in the state-of-the-art open-loop design, the high-pressure oil is discharged to the tank at zero pressure, wasting a significant amount of input work from the hydraulic pump”.

Comment 1-8 The authors stated that work recovery reduced losses in multi-mode eC system (line 137). Did authors mean multi-regenerator instead of multi-mode?

Response: Thank you very much for your comment. We mean “multi-mode” here because ref. 27 is a “multi-mode eC cooling system” that can switch between the single-stage mode and the active-regeneration mode.

Comment 1-9 In line 216-220 disordered geometries are mentioned. An important disadvantage of such design is also highly inhomogeneous stress distribution, which further results in inhomogeneous eCE and thus poorer performance. This should be mentioned.

Response: Thank you very much for your suggestion. We agree. The “inhomogeneous stress

distribution” was added in the revised version.

Comment 1-10 It is mentioned in line 239 that in rotary systems air is usually used. I do not think it is necessary to use air, as also water can be used in a similar way. I would suggest omitting “(usually air)” from that sentence.

Response: Thank you very much for your comment. We have removed “usually air” based on your suggestion.

Comment 1-11 It is not clear why window-type air conditioning device require maximum temperature span of 59 K, if it usually work with a temperature span of about 20 K. this should be explained more in detail.

Response: Thank you very much for your comment. The temperature difference between the ambient air and indoor dew point is indeed 20 K. Additional temperature difference is needed to exchange heat. Here, a 5 K temperature difference is assumed on each side, and 30 K temperature span is needed from the eC system when it delivers useful cooling, i.e. 2.5 kW cooling power. Please note that the 59 K temperature span in Table 1 is different from the 30 K temperature span here, because 59 K is the maximum temperature span without any heat load. Since on the performance map, there is usually a linear correlation between the cooling power and temperature span (Fig. R1), the maximum temperature span can be considered twice the nominal temperature span.

Fig. R1. Comparison of maximum temperature span and nominal temperature span.

To address similar concerns, we added “assuming a linear correlation between the cooling power and temperature span on the performance map that has been previously reported^{13,22}” in the revised version.

Comment 1-12 It is mentioned (line 281) that the total input of air conditioner is less than 735 W. At which temperature span?

Response: Thank you very much for your comment. 735 W is the power of a standard 1 HP system, rated at the standard rating condition in Tab. 2 (ideal temperature span of ~ 20 K). In current vapor-compression systems, the evaporating temperature is usually ~ 5°C and the condensing temperature is usually ~ 45°C because 10 K additional temperature difference is needed in both the evaporator and the condenser. Thus the system temperature span is ~ 40 K.

Comment 1-13 In the last paragraph SMA thermal storage is discussed. Indeed, SMA has many advantages compared to traditional PCMs, but their latent heat is still quite smaller, which should be mentioned in the manuscript.

Response: Thank you very much for your comment. We agree. Although the volumetric latent heat of SMA is higher, the mass-based latent heat is indeed smaller. We added this in the revised version as “Although the mass-based latent heat of SMA is still limited” before “the volumetric thermal energy performance of SMA outperforms traditional paraffin wax-based materials, manifested by a much higher thermal conductivity in SMA than paraffin wax and its composite structure with copper or aluminum”.

Reviewer #2 (Remarks to the Author):

This is a review of the application research of elastocaloric cooling. The manuscript provides detailed state-of-the-art elastocaloric cooling and forecasts for the development of this technology. It helps to understand the elastocaloric application and inspires the researchers in this field. There are some minor shortcomings to be addressed to improve this review article.

Comment 2-1 The title of each section should be enlightened. Obviously, the present 2nd and 3rd paragraphs are not included in the Abstract. They should be in the section "Introduction". At the end of the Section "Introduction", the main contents or structure of this manuscript should be depicted clearly.

Response: Thank you very much for your suggestion. Originally we did have an introduction section heading. But this is contradictory to the journal's formatting policy. Therefore, the section heading of "Introduction" was deleted on purpose. You may refer to a published Perspective in this journal as an example: <https://www.nature.com/articles/s44172-022-00012-9>

Comment 2-2 At present, the already appeared prototypes of elastocaloric cooling almost used NiTi alloy. So, the title of this article should add "NiTi alloy". If you present other materials of elastocaloric application, in the text, they should be listed and the comparison in the coefficient of performance (COP), stress hysteresis, or anti-fatigue performance between them is necessary.

Response: Thank you very much for your suggestion. We agree with the reviewer that current prototypes were based on NiTi alloy. However, this is because NiTi is the best available material from the market as of today. NiTi may not be the best option in the future. For example, reducing stress and hysteresis by using NiTiCu or even copper-based SMAs is under development in a few research institutes and will be reported soon. Due to the same reason, these low-stress materials were discussed in the second paragraph of the "Solid-state elastocaloric refrigerants" section. Considering this paper is a forward-looking "Perspective" rather than a summary of the state-of-the-art, we prefer to eliminate the specific focus on "NiTi alloy" in the title.

And we do list these materials other than NiTi in the text, for example "...such as ternary TiNiCu or TiNiCo alloys are important (Table 1)" and "Materials with excessive latent heat but limited mechanical or fatigue performance, such as NiMnTiB...". The fundamental properties of these materials are given in Table 1, including temperature hysteresis (closely related to stress hysteresis and materials-level coefficient of performance). We do not specifically compare materials COP because it is temperature-dependent. For anti-fatigue performance, a fair comparison requires the same geometries, which is almost impractical because different materials were reported by different research institutes with different shapes and geometries. And since hysteresis is closely related to fatigue performance, we believe a comparison of the fundamental properties including hysteresis is sufficient for this Perspective here.

REVIEWERS' COMMENTS:

Reviewer #1 (Remarks to the Author):

Dear Authors, Dear Editor.

The authors correctly addressed my previous comments, and the manuscript can be accepted as it is.

Reviewer #2 (Remarks to the Author):

The authors have revised and improved their manuscript according to review comments. Now, I have no more questions.